# The Cell Envelope Stress Response of *Bacillus subtilis* towards Laspartomycin C

**DOI:** 10.3390/antibiotics9110729

**Published:** 2020-10-23

**Authors:** Angelika Diehl, Thomas M. Wood, Susanne Gebhard, Nathaniel I. Martin, Georg Fritz

**Affiliations:** 1LOEWE Centre for Synthetic Microbiology, Philipps-Universität Marburg, Hans-Meerwein-Strasse 6, 35032 Marburg, Germany; angelika.diehl@synmikro.uni-marburg.de; 2School of Molecular Sciences, The University of Western Australia, Crawley 6009, WA, Australia; 3Biological Chemistry Group, Institute of Biology Leiden, Leiden University, Sylviusweg72, 2333 BE Leiden, The Netherlands; t.m.wood@biology.leidenuniv.nl (T.M.W.); n.i.martin@biology.leidenuniv.nl (N.I.M.); 4Department of Chemical Biology & Drug Discovery, Utrecht Institute for Pharmaceutical Sciences, Utrecht University, Universiteitsweg 99, 3584 CG Utrecht, The Netherlands; 5Milner Centre for Evolution, Department of Biology and Biochemistry, University of Bath, Bath BA2 7AY, UK; sg844@bath.ac.uk

**Keywords:** laspartomycin C, friulimicin B, *Bacillus subtilis*, cell wall inhibition, stress response

## Abstract

Cell wall antibiotics are important tools in our fight against Gram-positive pathogens, but many strains become increasingly resistant against existing drugs. Laspartomycin C is a novel antibiotic that targets undecaprenyl phosphate (UP), a key intermediate in the lipid II cycle of cell wall biosynthesis. While laspartomycin C has been thoroughly examined biochemically, detailed knowledge about potential resistance mechanisms in bacteria is lacking. Here, we use reporter strains to monitor the activity of central resistance modules in the *Bacillus subtilis* cell envelope stress response network during laspartomycin C attack and determine the impact on the resistance of these modules using knock-out strains. In contrast to the closely related UP-binding antibiotic friulimicin B, which only activates ECF σ factor-controlled stress response modules, we find that laspartomycin C additionally triggers activation of stress response systems reacting to membrane perturbation and blockage of other lipid II cycle intermediates. Interestingly, none of the studied resistance genes conferred any kind of protection against laspartomycin C. While this appears promising for therapeutic use of laspartomycin C, it raises concerns that existing cell envelope stress response networks may already be poised for spontaneous development of resistance during prolonged or repeated exposure to this new antibiotic.

## 1. Introduction

The cell envelope is both the bacterium’s first line of defence against harmful substances and an essential structure to counteract the internal turgor pressure. As many antibiotics exploit the accessibility of the cell envelope, bacteria have evolved numerous countermeasures to protect themselves against these constant attacks. In Gram-positive bacteria in particular, the biosynthesis pathway of peptidoglycan (PG), the lipid II cycle, is a key antibiotic target [1,2] (Figure 1). As a cyclic pathway, the blockage at any point of the cycle is sufficient to bring PG synthesis to a halt [1], and consequently, all externally exposed cycle intermediates are inhibited by one or several antibiotics (Figure 1). While antibiotics targeting lipid II and undecaprenyl pyrophosphate (UPP) are widely used in therapeutic and commercial applications, e.g., vancomycin, nisin and bacitracin, currently no antibiotics targeting undecaprenyl phosphate (UP) are in use.

First hopes to exploit UP as a novel drug target against Gram-positive bacteria were based on friulimicin B—a naturally occurring cyclic lipopeptide produced by the actinomycete *Actinoplanes friuliensis* (Figure 2) [3]. However, clinical trials with friulimicin B were soon discontinued due to unfavourable pharmacokinetic properties [4]. While these properties can, in principle, be altered by introducing small chemical modifications in the polypeptide structure, this is difficult for friulimicin B, which is currently obtained naturally by extensive extraction from *A. friuliensis* [3]. Recently, laspartomycin C, another cyclic calcium-dependent lipopeptide, has been obtained by total chemical synthesis and was shown to also bind to UP as a drug target (Figure 2) [5,6]. In principle, the chemical synthesis of laspartomycin C allows for easier modification of the peptide’s pharmacokinetics, making this a promising route to develop a clinically relevant drug. However, while laspartomycin C has been examined in great detail biochemically [5,7,8], little is known about potential bacterial defence and stress response mechanisms against this novel antibiotic. Such an understanding is key to assess the risk of antibiotic resistance evolution and to develop early strategies to prevent cross resistance from stress responses triggered by any potential off-target binding in vivo.

*Bacillus subtilis* is a Gram-positive soil bacterium that is naturally exposed to a large range of antimicrobial peptides produced by competing environmental bacteria. The intricate cell envelope stress response (CESR) network protecting *B. subtilis* against these attacks (Figure 3) has become a model system for studying antibiotic resistance mechanisms in Gram-positive bacteria. Here, we set out to perform a comprehensive analysis of the *B. subtilis* CESR towards laspartomycin C and to compare it to the response against the structurally related friulimicin B [5]. Previously, friulimicin B was shown to induce the activity of several extra cytoplasmic function (ECF) σ factors involved in the CESR, with the greatest effect seen in σ^M^ and σ^V^ activation [9]. Under non-inducing conditions the anti-sigma factors YhdL/YhdK keep σ^M^ in an inactive state, but release it during cell envelope stress [10]. The free σ factor recruits RNA polymerase to specific target promoters to activate transcription of genes encoding general cell wall homeostatic mechanisms. The most noteworthy target of σ^M^ is *bcrC* [11], which encodes the UPP phosphatase BcrC—an integral component of the lipid II cycle that catalyzes the dephosphorylation of UPP to UP (Figure 1). Interestingly, during friulimicin B challenge, *bcrC* has been found to be one of the most highly induced genes [9], and it may play a role in friulimicin B resistance in combination with its function in replenishing the UP pool. 

Another module involved in the CESR is the two-component system LiaRS, which is kept inactive by LiaF under non-inducing conditions [12]. During antibiotic attack on the lipid II cycle, e.g., by bacitracin, or upon membrane perturbation, this inhibition ceases and LiaS phosphorylates the response regulator LiaR. In turn, LiaR activates 10 genes including its main targets *liaI* and *liaH* [12]. During cell envelope stress, the two encoded proteins, LiaI and LiaH, colocalize in small patches on the cell membrane and are thought to stabilize the membrane underneath holes in the peptidoglycan layer [13]. Although the Lia module has been shown to respond to a large range of different stressors [14,15,16], the actual trigger has not been identified yet. However, it is generally believed that the system indirectly senses the damage on the cell envelope rather than directly detecting the diverse range of stressors [13,17].

The third and last type of CESR modules in *B. subtilis* are the Bce-like systems BceRSAB [21], PsdRSAB [22] and ApeRSAB (formerly YxdRSAB) [23]. Bce-like systems are widespread in Firmicutes bacteria, including important pathogens such as *Staphylococcus aureus* and *Enterococcus faecalis* [24,25]. They consist of a two-component system that is functionally linked to an ABC transporter, with the best studied example being the BceRSAB module from *B. subtilis* (Figure 3). Here, the transport permease BceB directly interacts both with the ATP-binding protein BceA and with the histidine kinase BceS [26]. BceAB likely confers resistance by a target protection mechanism [27], detecting the antibiotic bacitracin in complex with its cellular target UPP and removing bacitracin from UPP under ATP hydrolysis [18].The activity of BceAB then triggers activation of the histidine kinase BceS [19,26], which phosphorylates the response regulator BceR, leading to transcription activation of *bceAB* [20].

While it is known that friulimicin B does not activate any resistance modules of the *Bacillus subtilis* CESR apart from σ^M^ and σ^V^ [9], no such research has been done for laspartomycin C. Here, we found that while laspartomycin C triggers induction of the σ^M^ response similar to friulimicin B, it also strongly activates other resistance modules involved in the CESR, leading us to investigate their role in potential laspartomycin resistance.

## 2. Results

### 2.1. Laspartomycin C Induces a Slightly Stronger Response of the σ^M^ Regulon than Friulimicin B

The UPP phosphatase BcrC catalyzes a key reaction in the lipid II cycle, and is therefore highly expressed during normal growth [21,28]. Upon blockage of UP by friulimicin B, the increased activity of σ^M^ leads to a further boost in *bcrC* expression [9], thus possibly contributing to friulimicin B resistance. Given that laspartomycin C also targets the UP pool, we reasoned that it may likewise trigger the induction of the σ^M^ response, leading to upregulation of *bcrC*. To compare the σ^M^ response towards the two antibiotics, we used a strain of *B. subtilis* W168 harbouring a genomically integrated luciferase reporter under the control of the P*_bcrC_* promoter [11]. When exponentially growing cells of this strain were challenged with either of the antibiotics, the P*_bcrC_* promoter was most active between 30 min and 1 h after antibiotic challenge (Figure 4a,b). Figure 4c shows the dose dependency of luciferase activity, monitored 30 min after antibiotic addition. Here it turned out that while the P*_bcrC_* promoter reached a plateau in activity at 3 μg/mL friulimicin B and beyond, laspartomycin C maximally activated the P*_bcrC_* promoter at the higher concentration of 5 μg/mL (Figure 4c). While this suggests that the σ^M^ response is marginally more sensitive to friulimicin B than laspartomycin C, the P*_bcrC_* response was slightly stronger for laspartomycin C (3.5-fold induction) compared to friulimicin B (2-fold induction).

It had been reported previously that *B. subtilis* was more sensitive to friulimicin C than laspartomycin B, with MIC values of 0.078 and 8 μg/mL, respectively [8,9]. To test whether this was also true for exponentially growing cells under the conditions in our reporter gene experiments, we next tested the effect of both antibiotics on cell growth following challenge in the exponential growth phase (Figure 4d,e). The results confirmed that friulimicin B was more potent than laspartomycin C, with 5 μg/mL friulimicin B completely inhibiting cell growth, while 5 μg/mL laspartomycin C only marginally affected cells (Figure 4d,e). Closer examination showed that a 50% reduction in growth after 10 h (IC_50_, used as proxy for MIC) was achieved at 1.6 μg/mL friulimicin B, which was significantly lower than the IC_50_ for laspartomycin C—7.3 μg/mL (Figure 4f).

For effective protection, it is vital that resistance modules sense and respond to an antibiotic challenge before significant damage accrues. However, as IC_50_ values can vary widely between antibiotics, the critical concentration at which an antibiotic challenge needs to be sensed varies just as much. Therefore, we reasoned that the sensitivity of the σ^M^ response should be considered relative to the IC_50_ of the respective antibiotic, allowing us to place the regulation into a better physiological context. For this, we normalized the P*_bcrC_* dose–response curves in Figure 4c to the IC_50_ values. Growth kinetics of the strain harbouring the P*_bcrC_* reporter are shown in Appendix A. This showed that laspartomycin C maximally activated the σ^M^ stress response at concentrations around the IC_50_ value, while full induction by friulimicin B required concentrations exceeding the IC_50_ at least 2-fold (Figure 5). As such, we conclude that laspartomycin C is actually the more potent inducer of σ^M^ under physiological relevant conditions, i.e., at antibiotic concentrations below the IC_50_ value.

### 2.2. Probing the Broader CESR against Laspartomycin C Shows Induction of the LiaFSR Module

We next wanted to test whether laspartomycin C also activates other CESR modules of *B. subtilis*. The Lia system responds to a broad range of cell envelope-perturbing agents as described above [29,30]. Given this broad inducer spectrum, it was surprising that the UP-binding antibiotic friulimicin B did not activate the Lia system [15]. Based on our observation that laspartomycin C was a more potent inducer of the σ^M^ response than friulimicin B, we wondered whether this antibiotic could elicit a Lia response, or whether the Lia system generally did not react to UP-binding compounds.

To gain insight into the response of the Lia system toward laspartomycin C, we monitored the activity of P*_liaI_*, the main target promoter of the LiaFRS sensing module [12], via a luciferase-reporter fusion integrated into the genome of *B. subtilis* W168. We assessed luciferase activity at a time point close to the maximal induction of P*_liaI_* (30 min after antibiotic challenge), and plotted these activities as a function of the respective IC_50_-normalized antibiotic concentration to directly analyse the physiologically relevant sensitivity of the Lia module (Figure 6). Growth kinetics of the strain harbouring the P*_liaI_* reporter are shown in Appendix A.

In agreement with earlier studies [9,15], friulimicin B did not activate the Lia response in our experiments (Figure 6). In contrast, laspartomycin C led to a 25-fold P*_liaI_* induction (maximal induction at 2xMIC: 80-fold). This shows that the Lia response can indeed be triggered by UP-binding antibiotics. The laspartomycin C response represents an intermediate induction of the Lia module, as the strongest inducer known so far—bacitracin—activates the promoter 100-fold around the IC_50_ in our setup (Appendix A). These results further suggest that the higher sensitivity towards laspartomycin C was not specific for just the σ^M^ resistance module but rather a more general phenomenon. Moreover, our result show that laspartomycin C, but not friulimicin B, can be sensed by several CESR modules.

### 2.3. Induction of Specific CESR Modules by Laspartomycin C

In contrast to the Lia response, the Bce-like modules of *B. subtilis* have a much more specific inducer spectrum, which made us wonder how these will respond to laspartomycin C [17]. The two best-understood modules, Bce and Psd, likely sense antibiotics in complex with their membrane-anchored target. So far, most of the Bce- and Psd-inducing antibiotics bind to the diphosphatic lipid carriers UPP and lipid II [22]. Given that laspartomycin C and friulimicin B block the monophosphatic UP molecule, it was unclear whether these antibiotics would be able to trigger activation of the Bce-like CESR modules. Since the Ape module is less understood, we restricted our analysis to the Bce and Psd modules and studied their activity via genomically integrated luciferase cassettes under the control of the P*_bceA_* and P*_psdA_* promoters in *B. subtilis* W168 using the previously described setup. Growth kinetics of the strains harbouring the P*_bceA_* and P*_psdA_* reporters are shown in Appendix A, respectively.

Dose–response curves 30 min after antibiotic challenge show that friulimicin B activated neither the Bce (Figure 7a) nor the Psd module (Figure 7b). Laspartomycin C, however, induced P*_bceA_* 25-fold at the IC_50_ (=maximal induction) (Figure 7a). While this response was weaker than the full induction of P*_bceA_* under bacitracin stress (~170-fold) [18], it still showed a pronounced activation. Similarly, the Psd module was activated 25-fold by laspartomycin C around IC_50_ (maximal induction: 30-fold) (Figure 7b). The observation that laspartomycin C, but not friulimicin B, activates these systems suggests that, while both antibiotics are able to bind to UP, only laspartomycin C can be recognized by the transporter and trigger its activity.

### 2.4. Acitvated Resistance Modules do not Protect against Laspartomycin C Attack

After establishing that laspartomycin C induces all three types of CESR modules in *B. subtilis*, we next asked whether the activated target genes conferred any protection against the antibiotic. Therefore, we determined the IC_50_ of laspartomycin C in wild-type *B. subtilis* W168, and in strains carrying deletions of each of the modules or their key target genes, i.e., Δ*bcrC*, Δ*liaIH*, Δ*bceRSAB* and Δ*psdRSAB* (Appendix A), using the methodology described above. Surprisingly, none of the tested deletion strains showed a significant reduction in the IC_50_ of laspartomycin C and friulimicin B (Student’s t test, *p* > 0.001) (Figure 8), suggesting that none of the induced genes actually contributed to protection of the cell against these antibiotics. Note that growth in the absence of the antibiotics was not affected by any of the deletions (Appendix A). Growth kinetics of all deletion strains challenged with laspartomycin C or friulimicin B are shown in Appendix A, respectively.

Given the complexity of the laspartomycin C stress response, it was possible that deletion of a single resistance determinant may not be sufficient to cause a detectable change in laspartomycin C sensitivity. A similar observation was made previously in the bacitracin stress response of *B. subtilis*, where the contribution of the Lia system to resistance was masked by the strong resistance mediated by BceAB, and a protective effect of Lia could only be observed when both *bceAB* and *liaIH* were deleted. To examine potential redundancy of the resistance modules during laspartomycin C challenge, we next tested the susceptibility of strains with deletions of two resistance modules combined. However, even in these double-deletion strains, no increase in susceptibility was detected (Student’s t test, *p* > 0.001) (Figure 8). This indicates that none of these resistance modules, even though strongly expressed, can protect *B. subtilis* against laspartomycin C.

## 3. Discussion

As a proxy for the potential of resistance development by the novel cyclic calcium-dependent lipopeptide laspartomycin C, we here assessed its ability to trigger elements of the CESR of *B. subtilis*. This was compared to the response to the better-characterized antibiotic friulimicin B, which shares similarities in chemical structure with laspartomycin C. Here, we found significant differences, as friulimicin B only triggered the induction of the σ^M^ component of the CESR, while laspartomycin C produced a stronger σ^M^ response and additionally activated both the Lia and Bce-like modules. Since the two antibiotics are closely related and bind the same target—UP—this was highly surprising.

### 3.1. Novel Clues on CESR Signal Perception

While the σ^M^ and Lia modules have been broadly referred to as “damage-sensing” systems [31], their specific triggers are currently unknown. Possible cues include the sensing of perturbations in the peptidoglycan layer, in the membrane [31], or an altered abundance of lipid II cycle intermediates [13] by some unknown mechanism. Since friulimicin B and laspartomycin C both bind to, and hence sequester, UP, it is likely that both antibiotics have similar effects on the abundance of the other lipid II cycle intermediates, i.e., decreased levels of UPP, lipid I and lipid II. The similarity in the σ^M^ response triggered by both antibiotics (2-fold induction by friulimicin B vs. 3.5-fold induction by laspartomycin C) might be reflective of such effects. This could be compatible with the σ^M^ module detecting changes in pool levels of lipid II cycle intermediates. Interestingly, however, we detected a strongly differential response of the Lia module toward the two antibiotics (no induction by friulimicin B vs. 25× induction by laspartomycin C). One might therefore argue that the Lia module is less likely to sense a reduction in lipid II cycle intermediates, as such a reduction should be very similar for both antibiotics. While we cannot provide a definitive answer to this question, the differential responses elicited by friulimicin B and laspartomycin C may be able to serve as novel tools to elucidate the physiological triggers of the Lia module and potentially of σ^M^.

In contrast to the indirect (damage-)sensing mechanism employed by the Lia and σ^M^ systems, the Bce-like modules are known to directly sense the antibiotic–target complexes, where the target is typically a diphosphatic lipid carrier (UPP or lipid II) [22]. Although it is still unknown which parts of the antibiotic–target complex these modules react to [18,22,32], individual modules specifically respond to a small set of antibiotics and often discriminate between very similar peptides. For example, of the two globular lantibiotics mersacidin and actagardine, only the latter induces the Psd module [22]. Recently, the two cannibalism toxins Sdp and Skf were also shown to induce the Bce and Psd modules [33]. Here, the membrane-anchored immunity protein SdpI was essential for induction of both modules, suggesting that the two toxins might be sensed in complex with SdpI. With laspartomycin C, we show here that the inducer spectrum of the Bce and Psd modules also extends to antibiotics targeting the monophosphatic lipid carrier UP. This unfolding promiscuity of compounds recognized by Bce-like systems may offer insights into the mode of action of these unique resistance transporters. The antibiotics appear to be only recognized in complex to their cellular target. However, this cellular target can apparently be either diphosphatic (UPP, lipid II) or monophosphatic (UP) lipid carriers, and even proteins (SdpI). This observation potentially extends the notion that Bce-like transporters act by target protection of cell wall synthesis [18,34], to target protection of the extracellular face of the membrane more generally, where the transporters are responsible for removing membrane- and cell wall-perturbing compounds. 

### 3.2. Differential Sensing of Friulimicin B and Laspartomycin C

While the calcium-dependent lipopeptide antibiotics friulimicin B and laspartomycin C share structural similarity, there are also a number of differences in the sidechains of the constituent amino acids and in the length and orientation of their lipid tails (Figure 2). Most notable among these differences is the presence of neutral Gly and d-*allo*-Thr residues at positions 4 and 9, respectively, in laspartomycin C, while these positions are filled by l-*threo*-3-methyl-aspartate (MeAsp) and 2*R*,3*R*-d-Dab, respectively, in friulimicin B. The crystal structure of laspartomycin C bound to geranyl phosphate indicates that residues 4 and 9 are solvent accessible [8]. Given that friulimicin B has acidic and basic residues at these positions, it may be possible that intermolecular salt bridges are formed between neighbouring friulimicin B–UP complexes. This in turn might lead to multimerization on the bacterial cell surface. Should friulimicin B form such multimeric structures, it may be that individual friulimicin B molecules become less accessible to the sensory machinery of resistance modules (Figure 9a). In contrast, given that laspartomycin C features neutral residues at positions 4 and 9, it may be less likely to form higher-order complexes. In this case, the more disperse laspartomycin C molecules might be more accessible to direct sensing by Bce-like modules (Figure 9b). Ongoing structural investigations are aimed at clarifying this hypothesis.

This model, however, only explains the differing responses towards the two antibiotics for resistance modules with a direct sensing mechanism. For damage-sensing modules, the differential responses are much more difficult to explain. As the trigger of σ^M^ is very controversial [35] and the induction of the P*_bcrC_* promoter by the two antibiotics did not show marked differences (<2-fold), we believe that strong conclusions about a differential σ^M^ response towards laspartomycin C and friulimicin B are not warranted based on our data. In contrast, the trigger of the Lia module is less hazy as it is generally believed to be either discrepancies in lipid carrier pool sizes or membrane perturbations [13,31]. As the former model is not in line with the differential Lia response towards friulimicin B and laspartomycin C, as described above, we favour a model in which the Lia module senses membrane perturbations. However, the exact nature of these perturbations remains enigmatic [13]. Within this model for Lia function, the observation that friulimicin B is not an inducer of the Lia response can be rationalised by predicting that friulimicin B would trigger less severe membrane perturbations than laspartomycin C. Such differences may arise, for instance, due to the longer and trans-configurated lipid tail of laspartomycin C, which creates a bulkier structure on the membrane surface than friulimicin B. This bulkiness might prevent membrane lipids from organizing in an orderly fashion and could lead to the accumulation of more fluid lipids around UP–laspartomycin C complexes, thereby preventing gaps in the membrane. This mode of action has previously been observed in the related cyclic calcium-dependent lipopeptide daptomycin, which also binds to a lipid in the cellular membrane—phosphatidylglycerol [36,37,38]. Due to its bulky lipid tail, fluid lipids cluster around daptomycin—a process that is further amplified by daptomycin multimerization [36]. This clustering has been shown to cause a depletion of fluid lipids in the remainder of the cellular membrane and increases overall membrane rigidity [38]. Since daptomycin also induces the Lia module, we propose that the depletion of fluid lipids from the membrane and/or the accompanied increase in membrane rigidity might be the molecular trigger for the induction of the LiaFSR signaling system [9,36]. A similar mechanism can be envisioned for laspartomycin C (Figure 9b), although future experiments will be needed to further corroborate such a model.

### 3.3. Lacking Antibiotic Protection is Widespread in Bce-like Resistance Modules and may Precede the Spontaneous Evolution of Resistance

One intriguing finding of this study was that even though laspartomycin C induces the expression of the Bce and Psd modules, none of these systems conferred detectable resistance against this novel antibiotic. Interestingly, similar observations have been made for other inducers of Bce-like resistance modules in *B. subtilis*. For instance, while the Bce module is most highly induced by both bacitracin and the lipid II-binding antibiotic mersacidin, it confers a high level of resistance against bacitracin (30-fold change in susceptibility) but only moderate resistance against mersacidin (4-fold change in susceptibility) [22]. Further, the aforementioned cannibalism toxins SdpC and SkfA are strong inducers of BceAB and PsdAB expression, but neither of the modules confers resistance against these toxins [33]. Similarly, the PsdRSAB module is induced by lipid II-binding antibiotics, such as actagardine and gallidermin [22], but despite actagardine being the strongest known inducer of *psdAB* expression, PsdAB does not confer any detectable actagardine resistance [22]. This indicates that the strength of induction of Bce-like modules by an antibiotic does not necessarily correlate with the level of protection conferred. 

Astoundingly, the same lack of protection against laspartomycin C, despite clear induction responses, was observed for every resistance module we tested in this study. Thus, it seems that the path to a fully functional resistance module is a two-step process: (1) sensing of the antibiotic leading to gene expression and (2) effectively counteracting the antibiotics. In the case of the Bce-like modules, both functions are presumably carried out by the transporters, BceAB or PsdAB [21,39,40]. We have shown previously that signaling is proportional to transporter activity [20], and thus signaling and resistance should also be directly coupled. However, the proposed target protection mechanism of BceAB action may provide some clues as to the observed discrepancy between gene expression and level of resistance. As discussed above, the physiological substrate for the transporter is the antibiotic in complex with its cellular target, and the energy from ATP hydrolysis is presumably used to break the antibiotic–target interactions and free the target [18,27]. For this process to lead to effective protection, BceAB activity has to result in an equilibrium shift towards free target that is sufficient to allow cell wall synthesis to continue. The freeing of the target therefore has to be fast relative to the renewed binding of the antibiotic. However, according to our current understanding of flux-dependent signaling, activation of the kinase will occur in proportion to ATP hydrolysis [19,20,26], irrespective of whether this results in effective freeing of the target. Conceptually, signaling should therefore be simpler to achieve, requiring only recognition of the substrate–target complex by the transporter, whereas resistance additionally requires suitable kinetic properties of the transporter to facilitate target protection. 

Given our findings, a potential route for the evolution of resistance against other/novel antibiotics by Bce-like modules may present itself. Considering the wide range of antibiotics that can be sensed by Bce-like modules it is relatively likely that a novel antibiotic could also induce these systems, as we observed here for laspartomycin C. As this already accomplishes the first step towards a functional resistance module, it is conceivable that continued selection pressure, for example through clinical use of a new antibiotic, would easily result in mutations that improve transport kinetics to the point where the antibiotic is effectively removed from its target and resistance is achieved. While experimental evolution of laspartomycin C resistance in *B. subtilis* was beyond the scope of this study, it will be interesting to test this in the future, should larger quantities of laspartomycin C become available.

Since Bce-like resistance modules are widespread in Firmicutes bacteria, including in important pathogens such as *S. aureus* and *E. faecalis*, the fact that laspartomycin C is able to induce these modules poses a considerable risk that resistance against this antibiotic might develop faster than for an antibiotic not already recognized as an inducer [24,25]. As such, further development of laspartomycin C as a clinical drug candidate must address these inducing properties and eliminate them if possible. More generally, testing a novel antibiotic for induction of known resistance systems may provide a fast, initial laboratory test for gauging the risk for developing resistance by adaptation from known resistance systems.

## 4. Materials and Methods

### 4.1. Bacterial Strains and Growth Conditions

All strains used in this study are listed in Table 1. *B. subtilis* was routinely grown in MOPS media [41] with added glucose (1.8% (*w*/*v*)) and tryptophan (0.05% (*w*/*v*)) at 37 °C and agitation at 220 rpm. Bacterial growth was monitored as optical density at a wavelength of 600 nm (OD_600_). Solid media contained 1% (*w*/*v*) agar. Selective media contained chloramphenicol (5 μg/mL), tetracycline (12.5 μg/mL) or kanamycin (5 μg/mL). 

### 4.2. Luciferase Reporter and IC_50_ Determination Assay

Stationary cultures were diluted 1:50 in fresh media and incubated for 5 h at 37 °C and 220 rpm. The cultures were subsequently diluted to OD_600_ 0.01 and loaded onto a black 96-well plate. Antibiotic dilutions were added and measurements (OD_600_ and luminescence) were taken every 10 min for 12 h in a CLARIOstar reader (BMG Labtech, Germany) at 37 °C. Lids were used to reduce evaporation. Cultures were agitated in between measurements in the corner well meandering mode at 300 rpm. All experiments were performed with the addition of 1.25 mM CaCl_2_. All experiments were performed in six biological replicates, but the P*_liaI_* promoter activity, which was measured in five replicates.

### 4.3. Data Processing

All data were analysed with custom scripts using Python. Measurements were smoothed using a median filter (window size = 3). Luminescence output was normalized to cell density by dividing each data point by its corresponding blank-corrected OD_600_ value (RLU/OD). Dose response was measured after one hour. The IC_50_ is defined as the minimal antibiotic concentration reducing the OD_600_ by 50% in comparison to unperturbed growth and was measured 10 h after antibiotic challenge. The true IC_50_ was estimated via a linear fit between the measured concentrations neighbouring the IC_50_.

## Figures and Tables

**Figure 1 antibiotics-09-00729-f001:**
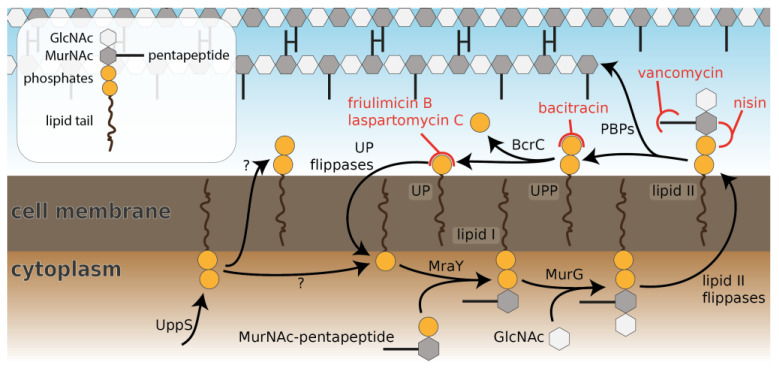
Schematic overview of the lipid II cycle of *Bacillus subtilis*, including lipid carrier molecules, involved enzymes and selected antibiotics that target cell wall synthesis. The cell wall precursor lipid II is assembled on the cytoplasmic side of the membrane and then flipped. After insertion of the GlcNAc-MurNAc pentapeptide into the growing peptidoglycan, the lipid carrier UPP is recycled via dephosphorylation by BcrC. The generated lipid carrier UP is subsequently flipped back to the cytoplasmic side of the membrane to begin a new cycle. The lipid carrier is supplied by UppS in the form of UPP on the cytoplasmic side of the membrane; it remains to be elucidated whether newly synthesized UPP is fed into the cycle by dephosphorylation on the cytoplasmic side of the membrane or by flipping. Example antibiotics that target lipid carrier pools on the periplasmic side of the membrane are indicated in red.

**Figure 2 antibiotics-09-00729-f002:**
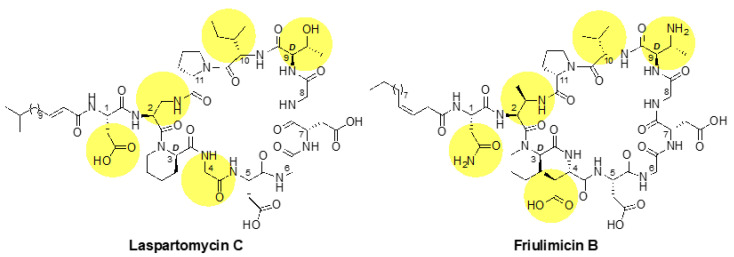
Structures of laspartomycin C and friulimicin B. The lipid tails of the two lipopeptides differ, as laspartomycin C bears a 15-carbon tail containing a *trans* alpha-beta unsaturated moiety while friulimicin B contains a 14-carbon *cis*-beta gamma unsaturated lipid. Other notable differences include the amino acids at positions 1, 2, 4, 9 and 10 (highlighted), which are Asp, diaminoproionic acid (Dap), Gly, d-*allo*-Thr, and Ile, respectively, in laspartomycin C and Asn, 2*S*,3*R*-diaminobutyric acid (Dab), l-*threo*-3-methyl-aspartate (MeAsp), 2*R*,3*R*-d-Dab and Val, respectively, in friulimicin B.

**Figure 3 antibiotics-09-00729-f003:**
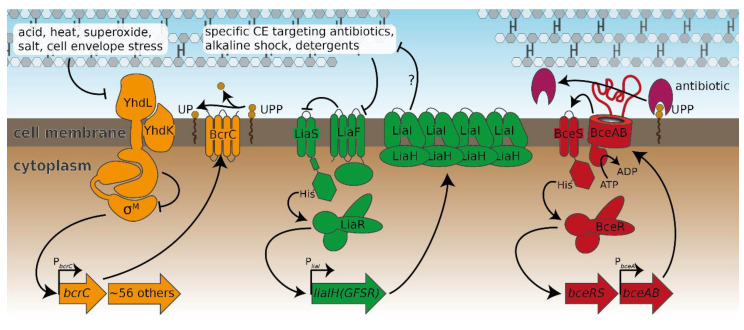
The three major cell envelope stress response modules of *B. subtilis* analysed in this study. The σ^M^ module (left) is kept inactive under non-inducing conditions by its anti-σ factors YhdL/YhdK [10]. Upon exposure to, e.g., acids or cell envelope stress, σ^M^ is released and free to guide RNA polymerase to its target promoters such as P*_bcrC_* [11]. The Lia module (middle) senses a plethora of external stresses through a yet-undetermined sensing mechanism. Upon stress sensing, the inhibition of the two-component system LiaS/LiaR by LiaF ceases [12], which subsequently allows increased expression of the genes involved in the signaling cascade as well as *liaI* and *liaH*. The Bce module (right) is shown as a representative of all bce-like modules of *B. subtilis*. Here, the ABC-transporter BceAB is thought to remove the antibiotic bacitracin from its target, UPP [18]. BceAB activity stimulates the two-component system BceS/BceR and thereby leads to the increased production of BceAB [19,20]. Regulation patterns are depicted with arrows; arrowheads and T-heads indicate activation and inhibition, respectively. Operons activated by the resistance modules are shown as thick arrows.

**Figure 4 antibiotics-09-00729-f004:**
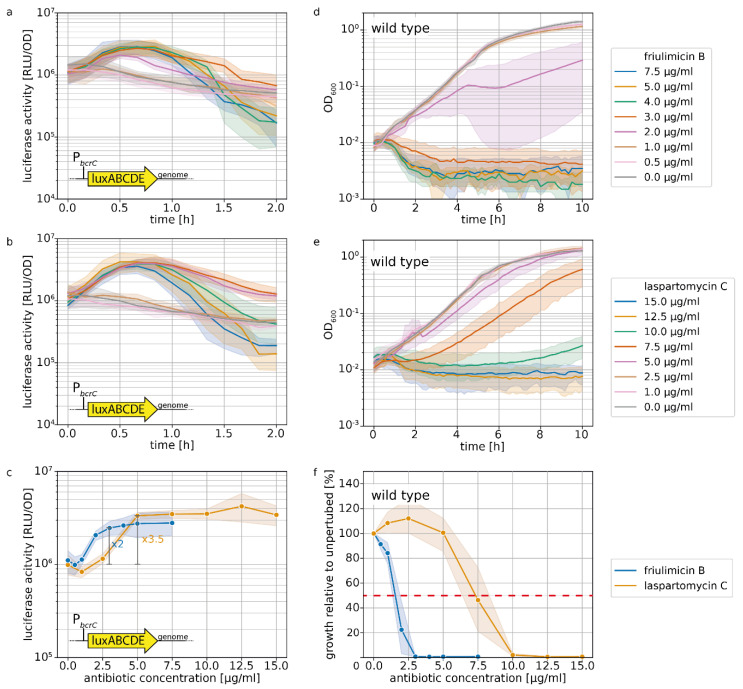
σ^M^ stress response as shown by *bcrC* promoter activation and growth during friulimicin B and laspartomycin C attack. (**a**,**b**) Expression of *bcrC* over time in dependency of different concentrations of friulimicin B (**a**) or laspartomycin C (**b**). (**c**) Dose dependency of the σ^M^ stress response 30 min after antibiotic challenge. Blue and orange lines depict the P*_bcrC_* promoter response generated by friulimicin B and laspartomycin C, respectively. The fold change over basal activity is shown at the beginning of the plateau. (**d**,**e**) Growth after antibiotic challenge with friulimicin B (**d**) or laspartomycin C (**e**). (**f**) Dose dependency of the growth relative to an unperturbed culture 10 h after antibiotic challenge. IC_50_ values were determined as the antibiotic concentration reducing bacterial growth by 50% (red dashed line), which was 1.6 and 7.3 μg/mL for friulimicin B and laspartomycin C, respectively. Shaded areas depict 95% confidence intervals.

**Figure 5 antibiotics-09-00729-f005:**
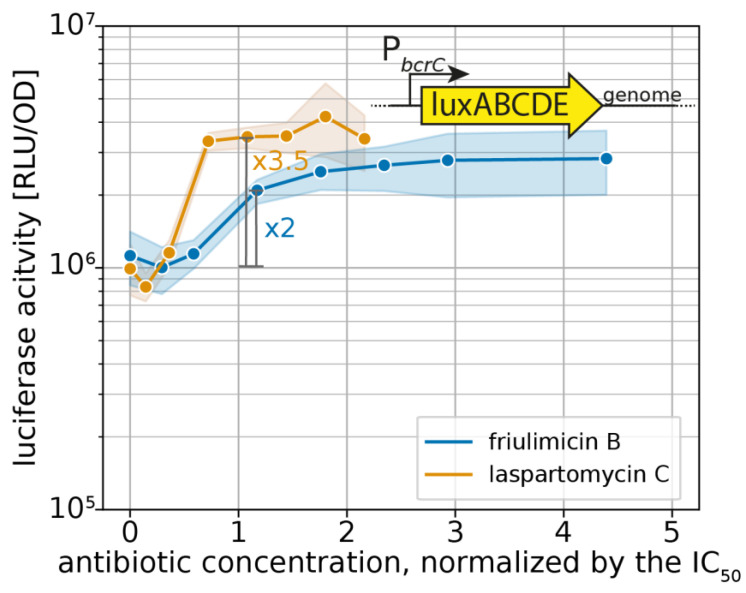
Sensitivity-normalized dose-dependent activation of the σ^M^ stress response as shown by *bcrC* promoter induction during friulimicin B and laspartomycin C attack. The antibiotic concentration is shown in relation to the IC_50_ of the respective antibiotic. The induction at the IC_50_ is indicated in grey, and the fold change is given. The induction at the IC_50_ is indicated in grey, and the fold change is given. Measurements were taken 30 min after antibiotic challenge. Shaded areas depict 95% confidence intervals.

**Figure 6 antibiotics-09-00729-f006:**
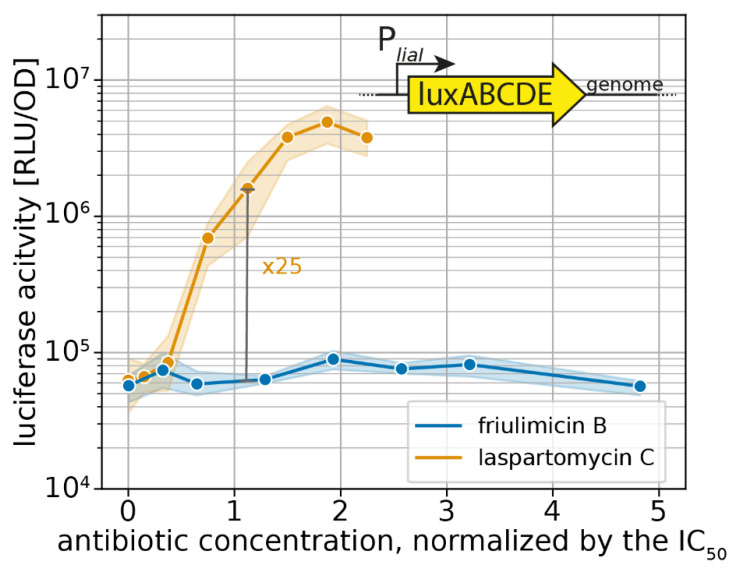
Sensitivity-normalized dose-dependent stress response of the Lia module as shown by *liaI* promoter activation during friulimicin B and laspartomycin C attack. The relative antibiotic concentration in respect of their IC_50_ is given. The induction at the IC_50_ is indicated in grey, and the fold change is given. Measurements were taken 30 min after antibiotic challenge. Shaded areas depict 95% confidence intervals.

**Figure 7 antibiotics-09-00729-f007:**
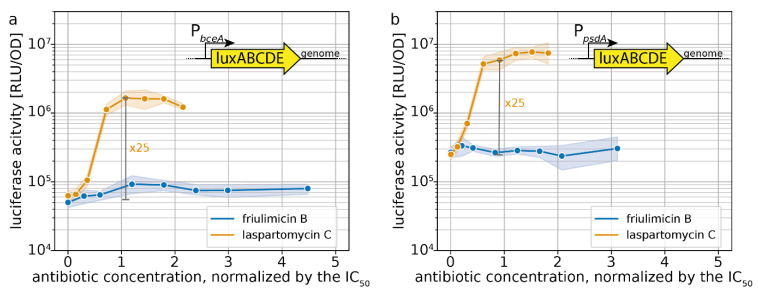
Sensitivity-normalized dose-dependent stress response of bce-like modules as shown by P*_bceA_* (**a**) and P*_psdA_* (**b**) promoter activation during friulimicin B and laspartomycin C attack. The relative antibiotic concentration in respect of their IC_50_ is given. The induction at the IC_50_ is indicated in grey, and the fold change is given. Measurements were taken 30 min after antibiotic challenge. Shaded areas depict 95% confidence intervals.

**Figure 8 antibiotics-09-00729-f008:**
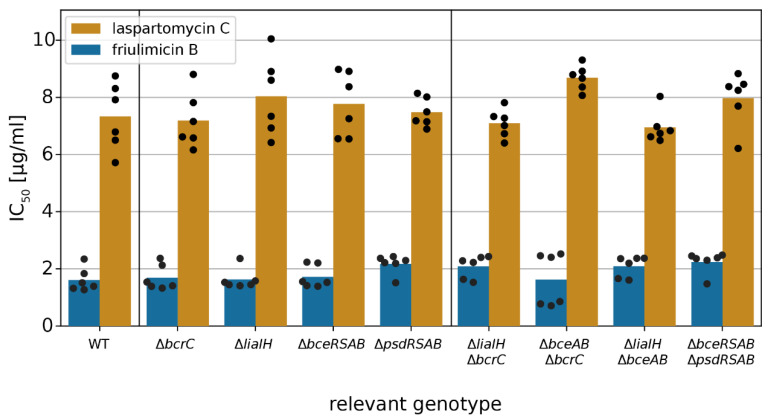
Impact of CESR modules on friulimicin B and laspartomycin C resistance. IC_50_ values were measured 10 h after antibiotic challenge in wild-type *B. subtilis* and deletion strains of CESR modules. The IC_50_ values of the wild-type strain were 1.6 and 7.3 μg/mL for friulimicin B and laspartomycin C, respectively. Bars represent the averaged IC_50_ over six biological replicates. Black dots show single replicates. Neither deletions of single CESR modules nor the combination of any two deletions changed the IC_50_ significantly (Student’s t-test: *p* > 0.001; unequal variance).

**Figure 9 antibiotics-09-00729-f009:**
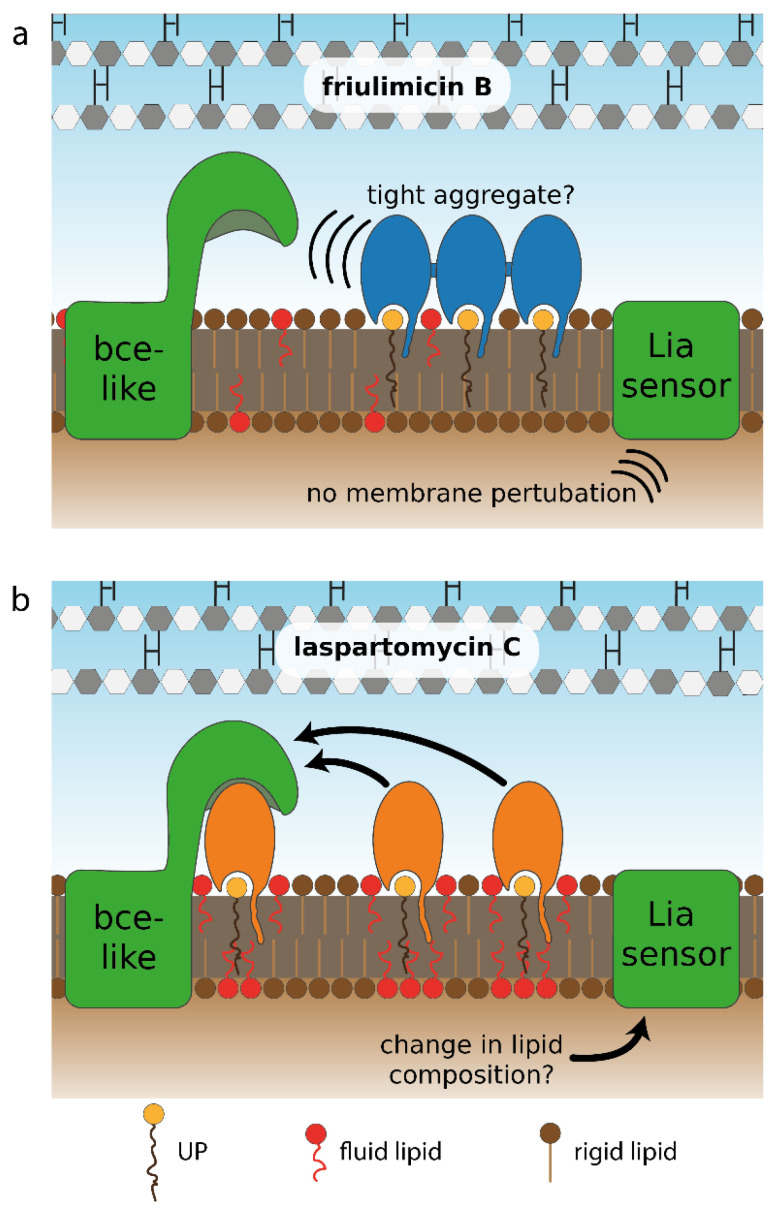
Hypothetical model for differential sensing of friulimicin B and laspartomycin C by Bce-like direct sensors and the LiaFSR damage sensor. (**a**) The proposed ionic interactions between neighbouring friulimicin B molecules may lead to formation of antibiotic–UP aggregates. Such tight packing of antibiotic–UP complexes could prevent the sensory complexes BceAB–BceRS and PsdAB–PsdRS from binding to the friulimicin B–UP complexes, thus interfering with the direct sensing of the Bce-like systems. (**b**) For laspartomycin C, in contrast, the bulkier lipid tail and the absence of salt-bridging amino acids in its peptide ring might suggest that it forms freely diffusing, non-aggregating complexes with UP. These complexes may be more amenable to interaction with the BceAB–BceRS and PsdAB–PsdRS sensory complexes, as suggested by the fact that BceAB detects both the antibiotic and the lipid carrier in the bacitracin–UPP complex [18]. Likewise, the bulkier lipid tail of laspartomycin C might trigger changes in membrane lipid composition, e.g., by depleting fluid lipids from other parts of the membrane, serving as potential trigger for the induction of the LiaFSR system. Such lipid re-arrangements could be absent for friulimicin B, when present in the tightly aggregated form as in (**a**), serving as a potential explanation why friulimicin B does not activate the LiaFSR system.

**Table 1 antibiotics-09-00729-t001:** Strains used in this study.

*B. subtilis* Strain	Source/Reference
W168 trpC2	Laboratory stock
W168 P*_bcrC_*-lux	[31]
W168 P*_liaI_*-lux	[31]
W168 P*_bceA_*-lux	[31]
W168 P*_psdA_*-lux	[33]
W168 *bcrC*::kan	[31]
W168 Δ*liaIH*	[31]
W168 Δ*bceRSAB*	[20]
W168 Δ*psdRSAB*	Intermediate strain to produce TMB1518 [42]
W168 Δ*liaIH bceAB*::kan	[31]
W168 Δ*liaIH bcrC*::tet	[31]
W168 *bceAB*::kan *bcrC*::tet P*_liaI_*-lux	This work

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
