# Peer review of "The Cell Envelope Stress Response of Bacillus subtilis towards Laspartomycin C"

_antibiotics, 2020, doi:10.3390/antibiotics9110729_

Round 1
Reviewer 1 Report
The manuscript by Diehl et al. describes the effect of Laspartomycin C antibiotic on bacterial cell wall using B. subtilis as a model organism. The comparison of Laspartomycin C with friulimicin B is interesting, though not highlighted in abstract. Use of multiple reporter strains has made the observations clear and informative. Overall the study is well designed, though some queries need to be addressed.
- The images need substantial improvement for their quality and clarity.
- the titles of results are not specific but more general, please consider rewriting the titles that clearly indicate the results of that particular section
- The data representation is though technically correct, but seems very confusing, without any error bars and statistical calculations. Image quality makes it even more difficult to understand the differences.
- The authors assume to utilize the predefined resistance modules of reporter strains. Did the authors try to generate resistant mutants against friulimicin B or Laspartomycin C? If yes, was there any cross-resistance between the two antibiotics?
Author Response
We thank the reviewer for her/his positive evaluation and thoughtful comments. Below please find our point-by-point responses:
>The images need substantial improvement for their quality and clarity.
We agree that the figure quality was poor in the converted pdf file. In the final submission, we supply high-quality vector graphics (in pdf format), which will hopefully solve this problem.
>the titles of results are not specific but more general, please consider rewriting the titles that clearly indicate the results of that particular section
We have corrected the titles for clarity, e.g. in line 228.
>The data representation is though technically correct, but seems very confusing, without any error bars and statistical calculations. Image quality makes it even more difficult to understand the differences.
It seems that due to the low image quality in the converted pdf version, the reviewer was unable to perceive the shaded areas around each curve, which indicate the 95% confidence levels for each measurement (as indicated in the figure captions). We believe that with the improved figure quality in the revised version this will be clear.
>The authors assume to utilize the predefined resistance modules of reporter strains. Did the authors try to generate resistant mutants against friulimicin B or Laspartomycin C? If yes, was there any cross-resistance between the two antibiotics?
So far we did not generate resistant mutants against laspartomycin C, but we are grateful for this suggestion and will consider these experiments in our future work.
Reviewer 2 Report
The manuscript titled “The Cell Envelope Stress Response of Bacillus subtilis towards Laspartomycin C” describes a thorough study aimed at identifying differences in the mechanism of action between Laspartomycin C and the most known lipopeptide antibiotic Friulimicin B.
Overall, the manuscript is well written. The authors widely describe the topic in the introduction section. The aim of the work is clear and the authors used very well the sections “Results” and “Discussion”. In particular, I really appreciated the paragraph 3.2.
Results are not conclusive, and much work should be made in the future. However, this manuscript opens the way for future studies by highlighting the peculiar mechanism of Laspartomycin C.
I only suggest some minor changes to authors:
- Figures 1, 3 and 9 are out of focus. Please improve the quality.
- Some typos are present along the manuscript (i.e. line 86, remove “of the”; line 216, replace “lead” with “led”; line 426, it looks lacking something.
Author Response
We thank the reviewer for the positive evaluation of our manuscript. Please find our point-by-point responses below:
I only suggest some minor changes to authors:
>Figures 1, 3 and 9 are out of focus. Please improve the quality.
We uploaded high-res pdf versions of our figures, which will hopefully solve these issues.
>Some typos are present along the manuscript (i.e. line 86, remove “of the”; line 216, replace “lead” with “led”; line 426, it looks lacking something.
Thanks for spotting these typos! They are now corrected.